# Classification, Localization and Quantization of Eddy Current Detection Defects in CFRP Based on EDC-YOLO

**DOI:** 10.3390/s24206753

**Published:** 2024-10-21

**Authors:** Rongyan Wen, Chongcong Tao, Hongli Ji, Jinhao Qiu

**Affiliations:** College of Aerospace Engineering, Nanjing University of Aeronautics and Astronautics, Nanjing 210000, China; w9ymyw@nuaa.edu.cn (R.W.);

**Keywords:** carbon fiber-reinforced plastic defects, eddy current nondestructive testing, classification, localization and quantization, improved YOLOv8 target detection methods

## Abstract

The accurate detection and quantification of defects is vital for the effectiveness of the eddy current nondestructive testing (ECNDT) of carbon fiber-reinforced plastic (CFRP) materials. This study investigates the identification and measurement of three common CFRP defects—cracks, delamination, and low-velocity impact damage—by employing the You Only Look Once (YOLO) model and an improved Eddy Current YOLO (EDC-YOLO) model. YOLO’s limitations in detecting multi-scale features are addressed through the integration of Transformer-based self-attention mechanisms and deformable convolutional sub-modules, with additional global feature extraction via CBAM. By leveraging the Wise-IoU loss function, the model performance is further enhanced, leading to a 4.4% increase in the mAP50 for defect detection. EDC-YOLO proves to be effective for defect identification and quantification in industrial inspections, providing detailed insights, such as the correlation between the impact damage size and energy levels.

## 1. Introduction

Composite materials, formed by combining two or more materials with different properties through physical or chemical methods, often exhibit unique characteristics unattainable by single materials. These composites are extensively utilized in aerospace, automotive applications, industrial manufacturing, new energy sectors, and more [1]. Carbon fiber composites, in particular, are highly valued for their high strength, light weight, excellent electrical conductivity, and superior corrosion and fatigue resistance. They play critical roles in aircraft fuselage, wing structures, spacecraft, satellites, ship components, high-performance racing cars, electric vehicle power battery shells, driveshafts, wind turbine blades, bridges, and seismic structures [2,3,4]. During the fabrication, service, and maintenance of complex-shaped CFRP structural components, defects such as cracks, delamination, porosity, and impacts may occur. These defects can directly impair the mechanical properties of the structures and pose significant safety hazards [1,5,6,7,8]. Consequently, studying nondestructive testing techniques for CFRP structural defects is essential. Eddy current nondestructive testing has garnered considerable attention due to its advantages, such as the absence of a coupling medium and the simplicity of the equipment [8,9,10]. Currently, the classification and identification of CFRP defects largely depends on manual efforts involving comparison and marking [1,11]. Meanwhile, the advent of deep learning, machine vision, and other advanced methods has significantly mitigated the drawbacks of manual defect identification, such as high subjectivity and labor costs [1,9,11,12]. For instance, Wang et al. employed the YOLOv8 model to classify, predict, and identify various types of cracks, like crescent fatigue cracks and web cracks, in rails. Their results indicate that the model can effectively classify and locate defect targets [13]. Similarly, Huang et al. demonstrated that an improved lightweight YOLOv5 network can quickly and accurately identify defect targets on steel surfaces [14]. Faiyaz et al. used a machine learning approach to detect defects in high-altitude transmission tower insulators, achieving accuracy of 93.5% and a speed of 58.2 frames per second [15]. Moreover, machine vision techniques have been widely researched and applied in intelligent agriculture, pavement defect identification, defect detection in aluminum core materials and fiber optic sensors, and multilayer structure delamination and damage detection [16,17,18,19,20]. The YOLOv8 model, in particular, has become increasingly important in industrial automation, intelligent transportation, medical image analysis, autonomous driving, and augmented reality due to its high accuracy, fast inference speed, and excellent generalization performance [21,22,23,24,25,26,27,28]. Liu et al. reported that using the YOLO algorithm for lightweight prohibited item detection improved the average accuracy by 4.98% compared to YOLOv4 [29]. Ma et al. showed that YOLO effectively recognized broken wire signals in concrete cylindrical pipes [30].

To the best of the authors’ knowledge, the current CFRP defect eddy current detection methods still require substantial manual involvement for the classification, identification, target detection, and quantification of defects. This reliance results in low accuracy, high subjectivity, and excessive labor costs. In addition, traditional machine learning methods are limited to handling simple CFRP defect classification tasks. In other words, traditional CFRP defect detection methods typically analyze eddy current inspection images containing only one defect type, such as cracks, delamination, or impact damage. Simple defective image classification tasks can usually be handled by tools such as perceptrons, support vector machines, and convolutional neural networks. However, in real-world manufacturing and service processes, CFRP structural components often exhibit multiple defects simultaneously, leading to diverse defect types within a single inspection image. Traditional image classification methods struggle to address this complexity, resulting in issues like dominant defects overshadowing other damages or causing misclassifications. Moreover, conventional defect classification lacks high-level information, such as the position information of the defect and the defect area’s size—crucial factors for production control and maintenance execution. Regarding the richer and more complex information embedded in CFRP defect detection results, i.e., the acquisition of the defect location and defect size information, it is more difficult to analyze this using traditional machine learning methods. Therefore, we need an advanced computer vision model that can solve complex tasks involving classification, multi-target detection, localization, and quantization.

To address the challenge of identifying, localizing, and quantifying multiple targets in eddy current detection images of CFRP defects, this paper employs the YOLOv8n model. The YOLOv8n model can correctly classify all three types of CFRP defect damage, which is a huge improvement over the Transformer architecture classification. This method leverages a deep neural network structure to extract high-level semantic information about multiple defects within a single image. It utilizes a pyramid-shaped network structure to gather feature information from multiple layers and employs pre-selected boxes to output both the class probabilities and classifications of multiple defect targets. To overcome problems such as the fixed shape of sensory fields in traditional convolutional networks, a deformable convolutional network is used in YOLOv8 to increase the feature recognition ability for irregular CFRP defects in CFRP. For the large-scale or even global eddy current detection features of CFRP defects, this study combines the advantages of the Swin-Transformer structure’s high efficiency in parallel computation and its ability to capture long-distance feature information to solve the problems of large areas of defect damage distribution. Finally, to improve the overlap between the prediction boxes and the ground truth boxes, the model uses a WIoU loss function to measure the size of the model error. This model is called Eddy Current YOLO (EDC-YOLO). The results demonstrate that EDC-YOLO excels in the classification, identification, localization, and quantification of carbon fiber-reinforced polymer (CFRP) eddy current detection defects. Compared to YOLOv8n, EDC-YOLO’s mAP50 performance for single-defect target detection is enhanced by 4.4% and the mAP50-90 is improved by 4.6%. Additionally, the prediction box loss during training is reduced by 57%, the precision is increased by 3.3%, and the recall by 3.4%. The integration of an improved module further augments EDC-YOLO’s capabilities in identifying and localizing cracks, delamination, and impact damage defects.

In practical scenarios, multiple types and quantities of CFRP defects often coexist within a single eddy current inspection image, necessitating simultaneous and accurate identification, localization, and quantification. By simulating the coexistence of multiple defects, this study demonstrates EDC-YOLO’s proficiency in multi-target detection, significantly enhancing the efficiency of CFRP eddy current inspections. Unlike traditional CFRP defect quantification processes that require complex post-processing and various data algorithms to extract and calculate the defect areas, EDC-YOLO also directly quantifies the defect feature areas by extracting the geometric information from the model’s output prediction box. For instance, in the case of CFRP impact damage defects, EDC-YOLO directly outputs defect area information, illustrating that the damage area increases with higher impact energy.

This paper is structured as follows. Section 2 introduces the eddy current detection system and examines three typical CFRP defects using eddy current nondestructive testing. Section 3 addresses a simple three-classification problem using the standard YOLOv8 model. Finally, Section 4 applies the enhanced EC-YOLO model to more complex tasks involving the identification, localization, and quantification of defects.

## 2. CFRP Eddy Current Inspection System with Defect Dataset

### 2.1. CFRP Eddy Current Detection System

The CFRP ECNDT system constructed in this study comprises a high-sensitivity nine-grid eddy current probe, a Stanford SR-844, Sunnyvale, CA 94089, USA, lock-in amplifier, a Pioneer, 301 Kaki Bukit Ave 1, Shun Li Industrial Park, Singapore 416085, YX1715A DC power supply, an Agilent, 5301 Stevens Creek BlvdSanta Clara, CA 95051 United States, 33220A 20 MHz function/arbitrary waveform generator, a ZXC motion controller, an NI, Austin, TX 78759-3504, USA, PCI-6251 data acquisition card, an industrial computer, and a motion testing platform. The high-precision nine-cell eddy current detection probe is depicted in Figure 1.

A signal generator produces a sinusoidal signal with an excitation frequency of 1 MHz and an amplitude of 1 Vpp, which is loaded into the excitation coil. The detection coil senses the secondary magnetic field generated by the induced eddy currents in the CFRP. The resultant sinusoidal detection voltage signal is then transmitted to the lock-in amplifier to extract the real and imaginary components of the detection voltage signal. The data acquisition card collects this output voltage information and sends it to the industrial computer for data processing, image display, and storage.

In the software aspect of the system, LabVIEW 2016 is utilized to automate the CFRP eddy current inspection process. The automated system commands the motion control platform, containing CFRP specimens, to perform two-dimensional round-trip motions. During the inspection, the high-precision nine-grid eddy current inspection probe is fixed at a specific height above the motion platform. CFRP test pieces with various defects are placed and secured on the motion platform, and the motion controller drives the platform in the X-Y plane. The detection area is set to 50 × 40 mm^2^, with the data acquisition card collecting 8 data points per 1 mm of X-direction movement, and the step size in the Y-direction is set to 1 mm. The scan starts above a defect-free area, requiring the lock-in amplifier’s output signal to be zeroed as the reference signal voltage.

### 2.2. CFRP Defect Detection Images

To generate eddy current inspection images of CFRP defects, CFRP specimens with cracks, delamination, and impact damage were prepared.

#### 2.2.1. Crack Damage

Specimens with crack defects were created by stacking 16 layers of carbon fiber prepreg fabric, each 0.125 mm thick, with varying orientations, and curing them by heating. The stacking sequence from top to bottom was ${\left[45/0/-45/90\right]}_{2s}$. Groove cracks of different geometries and orientations were milled onto the surfaces of these CFRP specimens, as detailed in the accompanying table. The prepared specimens and their respective crack parameters are shown in the subsequent figures and tables.

Figure 2 illustrates Specimen 1, a defective CFRP specimen containing cracks of varying sizes but identical orientations. The specific dimensional parameters are provided in Table 1.

Specimens 2 and 3 were processed similarly but with different orientation angles and geometries to capture diverse crack damage information, as shown in Figure 3.

The layup sequence for these specimens matched that of specimen 1. Cracks of different directions and sizes were milled onto their surfaces, with the dimensions listed in Table 2 and Table 3.

#### 2.2.2. Impact Damage

To simulate impact defects, the drop hammer impact method was employed, mimicking collisions and falling objects. The impact head used was spherical, with a diameter of 15 mm and a weight of 1.5 kg.

Different magnitudes of impact energy were generated by adjusting the height of the spherical impact head to act on the surface of the CFRP specimen. This study conducted eddy current detection experiments on impact damage with varying impact energies of 2.5 J, 3.5 J, 4 J, 5 J, 5.5 J, and 6 J. The layup sequence of the CFRP multidirectional plate was $[{45/0/-45/90]}_{2s}$, with each unidirectional layer having a thickness of 0.125 mm, resulting in a total size of 250 × 250 × 2 mm. The crack dimensions were 16 mm in length, 0.5 mm in depth, and 1 mm in width. The scanning area for eddy current detection was set to 50 × 40 mm^2^. Prior to conducting the experiment, the output signal voltage of the lock-in amplifier was calibrated to be close to zero as the reference signal. The defective region was then scanned using a C-scan, as illustrated in Figure 4.

#### 2.2.3. Delamination Damage

To simulate the common form of delamination damage in CFRP, insulating shims were embedded between the layers. Materials such as PTFE and release paper were inserted at specific positions and depths during the fabrication of the CFRP specimens, as shown in Figure 5.

After high-temperature curing, these materials isolated the electrical connection between the upper and lower layers to mimic the effect of delamination damage. The parameters of the CFRP specimens containing delamination damage are listed in the accompanying Table 4.

#### 2.2.4. Dataset

For the CFRP specimens with the three aforementioned defects, a high-precision, high-sensitivity, nine-grid eddy current probe was employed for detection. The detection results of the CFRP eddy current inspection system for these defects were captured and analyzed. Meanwhile, the number of training samples is critical for the neural network training process. A robust training set, rich in sample variety, enhances the generalization ability and robustness of deep network models. Deep neural networks can learn essential features and descriptive patterns from a diverse set of training samples, rather than merely memorizing data details. Data containing the same high-level semantic information but differing in appearance should be recognizable by the deep network. Thus, simple deformation operations on raw images containing high-level semantic information can be beneficial for network training. Such simple transformation operations include random cropping, random flipping, random scaling, luminance transformation, and random color transformation, as illustrated in Figure 6.

Using the above image augmentation methods, a dataset of 2131 images of three defects was created. This dataset was divided into training, validation, and test sets at the ratio of 7:2:1.

## 3. Classification of YOLOv8 Model Based on Magnitude Images

The defect feature image obtained from the eddy current NDT of CFRP defects includes both the amplitude image and the phase image of the output voltage signal. These two images represent the characteristic quantities of the sinusoidal voltage signal’s amplitude and phase, respectively. Traditionally, the analysis of defect characteristics involves combining these two images. However, recent developments have shown that some advanced algorithms can achieve high accuracy in classifying the three types of defects—cracks, delamination, and impact damage—using only the amplitude image of CFRP defect detection.

### 3.1. Transformer Model and YOLOv8n-cls Model

To investigate the effectiveness of using only amplitude images for CFRP defect classification in eddy current detection, this study employed the Vi-T, SWT, and YOLOv8 classification models on an amplitude image dataset for the CFRP defect multiclassification task. Specifically, the lightweight YOLOv8 classification model YOLOv8n-cls was utilized for the triple classification task of CFRP defects.

#### 3.1.1. Transformer Model

The Vi-T, PRI-SWT, and SWT deep neural networks, based on the Transformer architecture, play a crucial role in vision task processing. Among them, Swin-Transformer (SWT) is particularly noteworthy due to its low computational cost and high accuracy, making it ideal for various engineering scenarios. Developed by Microsoft Research in 2021, SWT is a deep network model designed for high-resolution images with dense visual tasks. It addresses the issue of the quadratic growth in Vi-T’s computational complexity with the image resolution and balances the trade-off between high precision and high speed.

The SWT model mitigates the computational pressure by segmenting high-resolution input images into stacked windows and calculating the self-attention scores and relative positional encoding within each window. To ensure information exchange between different windows, the model shifts the windows and computes the attention scores and feature maps for these displaced windows. The basic components of the SWT model are illustrated in Figure 7. The model comprises several sub-modules, including Patch Embedding, Patch Merging, and the Swin-Transformer Block (SWTB). The SWTB module consists of two parts: window attention calculation and displacement window calculation. By progressively decreasing the feature image resolution through the Patch Merging layer, the SWT model effectively detects both large and small targets by increasing the model’s receptive field.

#### 3.1.2. YOLOv8 Model

YOLOv8, as shown in Figure 8, introduced by Ultralytics in 2023, is a state-of-the-art (SOTA) model for image classification, object detection, image segmentation, and other vision tasks. The visual performance of the model is significantly enhanced by replacing the C3 sub-module with the C2f sub-module and restructuring the network header layer.

The YOLO model network consists of three main components: the backbone, neck, and head. The backbone handles feature extraction, and its ability to extract advanced semantic features is improved while reducing the model parameters through the C2f sub-module. The Convolution–Normalization–Activation (CBS) module increases the number of feature maps and reduces their resolution, creating a “pyramid” structure in the backbone network. The Spatial Pyramid Pooling Fast (SPPF) module concatenates the pooling results of various scales, enhancing the spatial multi-resolution capabilities of the YOLO model.

The neck component fuses multi-scale features, combining feature maps from different backbone stages with those in the neck to boost the feature representation. YOLO further increases the feature map resolution using an inverse convolutional layer.

The head component handles the final classification and target detection tasks. It includes the detection head, which predicts the anchor boxes, and the classification head, which handles classification. The detection head employs the Distribution Focal Loss (DFL) module to transform the input tensor and learn the predicted probability distribution in tasks like target detection, enhancing the YOLO model’s understanding of the prediction uncertainty.

### 3.2. Classification Analysis

The YOLOv8n-cls model, designed for lightweight performance, consisted of 99 layers, 1.44 million parameters, and 4.3 billion FLOPs. The configurations included a batch size of 32, 50 epochs, and an image size uniformly set to 224. The model employed the AdamW optimizer with a learning rate of 0.000714. Additionally, it utilized the CFRP defect eddy current NDT magnitude image dataset processed as in the previous section. The dataset comprised 1277 training images for crack, delamination, and impact damage, along with 854 validation images. The resulting multiclassification confusion matrix is displayed in Figure 9.

Here, label 1 represents crack defects, label 2 indicates delamination, and label 3 denotes impact damage defects. The definitions of the multiclassification indicators used in this paper are as follows.

Macro-P:(1)$$Marco-p=\frac{1}{n}\sum_{i=1}^{n}P_{i}$$

Macro-R:(2)$$Marco-p=\frac{1}{n}\sum_{i=1}^{n}R_{i}$$

Macro-F1:(3)$$Marco-F1=\frac{2}{\frac{1}{Marco-p}+\frac{1}{Marco-R}}=\frac{2\times Marco-p\times Marco-R}{Marco-p+Marco-R}$$

Kappa coefficient: This metric is used to assess consistency; higher Kappa values generally imply higher classification accuracy.
(4)$$Kappa=\frac{P_{0}-P_{e}}{1-P_{e}},P_{0}=\frac{\sum_{i}h_{ii}}{\sum_{i,j}h_{ij}},P_{0}=\frac{\sum_{i}{\left(\sum_{j}h_{ii}\right)}^{2}}{{\left(\sum_{i,j}h_{ii}\right)}^{2}}$$
where the precision rate $P_{i}$ in each category denotes the ratio of the diagonal element in the confusion matrix to the sum of the elements in the column of that element. The recall $R_{i}$ for each category is the sum of the elements in the row where the corresponding element is located. Based on the definitions of the multiclassification metrics macro-P, macro-R, macro-F, and the Kappa coefficient, the values of the classification model are presented in the following Table 5.

The results demonstrate that the classification accuracy of the SWT model can reach up to 96% when using only magnitude images. This indicates that the CFRP defect eddy current detection magnitude images contain rich defect damage information. Furthermore, an advanced deep network structure can ensure high accuracy in the classification task, without needing to fuse the phase information image results.

The experiment provides additional data related to the training process of the YOLOv8 model, as shown in Figure 10. Using the YOLOv8n-cls model, the accuracy has been maintained above 99% after the first five epochs, while the Swin-Transformer model’s classification accuracy rises to 90% after 20 epochs and stabilizes within the range of 94% to 95% after about 50 epochs.

Regarding the state-of-the-art (SOTA) YOLOv8 model, the use of magnitude images for classification tasks meets the engineering requirements, and it can completely and correctly classify the three types of CFRP defects: cracks, delamination, and impact damage. The YOLO model effectively extracts high-level feature information about local and global damage in CFRP eddy current detection defect images through high-dimensional feature extraction in the backbone network. It also utilizes the fusion of multi-scale feature maps during the processes of two-times downsampling and one-time upsampling in the neck, accurately accomplishing the classification task. Compared to the more advanced Swin-Transformer model, the YOLOv8 model exhibits higher accuracy.

## 4. Multi-Target Localization and Quantitation

The YOLO model demonstrates exceptional performance in the straightforward task of CFRP defect classification. However, in practical CFRP eddy current NDT engineering applications, defects are not always located at the center of the detection image, and their distribution is often random. Moreover, multiple damages can occur in a single detection, with signals from cracks, delamination, and other defects scattered across various locations in the eddy current detection image. These applications typically require the capability to identify defect targets, accurately pinpoint the defect locations, and initially quantify the defect damage area to predict the severity of the damage to the CFRP material.

To achieve sophisticated target detection, localization, and quantification, this paper introduces the EDC-YOLO model. This model addresses shortcomings such as the fixed receptive field morphology, low perceptual capability, insufficient noise suppression performance, and a lack of broad and global feature extraction capabilities. The modifications include the integration of deformable convolutional layers, the CBAM attention mechanism, and the Swin-Transformer module into the YOLOv8 model.

### 4.1. EDC-YOLO Critical Components

#### 4.1.1. Deformable Convolution

In eddy current detection images, CFRP defects vary significantly in size and shape, including circular, elliptical, and rhombus forms. The scale variability is also substantial; for instance, a nine-grid eddy current probe detecting a crack corresponds to a small circular region, while impact damage results in a larger rhombus-shaped region. Conventional convolutional structures struggle to capture these differential features in shape and scale effectively. To address this issue, this study draws on deformable convolutional (DConv) layers.

In conventional convolutional layers, the receptive field size and shape are fixed, making it challenging for the model to simultaneously and accurately extract the features of targets with varying scales and shapes. DConv layers can adaptively alter the shape of the convolutional kernel according to the target’s deformation, enabling the capture of targets with multiple scales and shapes and enhancing the model’s generalization ability.

The principle of DConv is illustrated in Figure 11, where each output feature map unit corresponding to the convolution kernel learns an additional orientation parameter. This allows the convolution kernel to morph into arbitrary shapes for the better tracking of geometric information such as target deformation. The process of calculating DConv can be expressed with the following equation:(5)$$y\left(P_{0}\right)=\sum_{P_{n}\in R}\omega (P_{n})\times x({P_{0}+P}_{n}+∆P_{n})$$
where y is the output value of the element point $P_{0}$ on the output feature map, R is the set of convolution kernel elements corresponding to this output point, $P_{n}$ is the element representing each of them, ω is the weight of the element, $∆P_{n}$ is the offset of this element, and x(·) is the input value of the corresponding point. The input values of the coordinate points are obtained using bilinear interpolation.

DConv runs as shown in Figure 12. The neural network is trained to characterize the position by parameter $P_{n}$, which allows the model to learn the deformation information of the target. Its dimension is [B,H,W,2N], N=k×k, and k is the size of the convolutional kernel.

#### 4.1.2. Convolutional Block Attention Module

To further improve the model’s performance in handling CFRP defect features of different scales and shapes, a Convolutional Block Attention Module (CBAM), which fuses channel attention and spatial attention, is employed in the model. This effectively enhances EDC-YOLO’s ability to sense and learn important channels and locations and improves the model’s defect detection capabilities.

The CBAM consists of a channel attention module and a spatial attention module wired in series, as shown in the following Figure 13.

A single input feature map of size H×W×C is fed into the global maximum pooling layer (GMP) and global average pooling layer (GAP), respectively, to obtain two feature maps of shape 1×1×C. These feature maps are then processed through a fully connected multilayer perceptron (MLP) for dimensionality reduction and restoration, summed, and activated to generate the channel attention feature map $M_{c}(F)$, i.e.,
(6)$$M_{c}\left(F\right)=\sigma \{MLP\left[AvgPool\left(F\right)\right]+MLP[MaxPool(F)]\}$$

Lastly, $M_{c}(F)$ is multiplied with the input feature map F to obtain the input feature map $F^{\prime }$ needed by the spatial attention module. Similarly, $F^{\prime }$ undergoes GAP and GMP operations in the channel dimension to obtain two feature maps of shape H×W×1. These maps are fused through a 7×7 convolutional kernel and activated to produce the spatial attention feature map $M_{s}(F)$:(7)$$M_{s}\left(F\right)=\sigma^{\prime }\{{Conv}^{7\times 7}(AvgPool\left(F^{\prime }\right);MaxPool(F^{\prime })\}$$
which is then multiplied with $F^{\prime }$ to obtain the final output feature map $F^{\prime \prime }$.

#### 4.1.3. Shift Window Transformer

CFRP defect eddy current detection features often exhibit different scales and shapes. While CNNs perform well on smaller and moderately sized defects, they are less effective for larger impact damage defects, where the defective region covers a large area. The Transformer architecture excels in capturing large-scale or global feature information, making it a suitable complement to CNNs. Combining the CNN and Transformer architectures allows the model to capture both local and global high-level semantic information, often resulting in better performance. To increase the computational efficiency and reduce the linear growth in the computational cost with the feature map resolution, the Swin-Transformer model divides the feature map into multiple shifted windows and computes the multi-head self-attention in each window. This allows the model to capture global feature information between different windows through displacement.

This study integrates Swin-Transformer into the head network of EDC-YOLO, forming the basic module of C3STR by replacing the bottleneck module in the C2f structure with the Swin-Transformer module, as shown in the Figure 14.

#### 4.1.4. WIoU Loss Function

The YOLOv8 model uses the Complete Intersection over Union (CIoU) as its loss function, which considers the overlap area of the prediction box and the target box, the distance between their centroids, and the aspect ratio, enhancing the robustness of the training process. The CIoU formula can be expressed as
(8)$$\begin{array}{c} L_{CIoU}=1-IoU\left(A,B\right)+\frac{\rho^{2}\left(A_{cent},B_{cent}\right)}{c^{2}}+\alpha \upsilon \\ \upsilon =\frac{4}{\pi^{2}}{\left(arctan\frac{w_{gt}}{h_{gt}}-arctan\frac{w}{h}\right)}^{2},\alpha =\frac{\upsilon }{\left(1-IoU\right)+\upsilon } \end{array}$$
where A,B denotes the prediction box and the target box, c is the diagonal length of the smallest enclosing box covering both the prediction and target boxes, $A_{cent},B_{cent}$ denotes the center point of the two boxes, and α,υ denotes the consideration of the aspect ratio.

Considering that the quality of the training samples is affected by disturbances in the CFRP defect production and detection processes, this study employs the Wise-IoU (WIoU) with a dynamic non-monotonic focal mechanism as the loss function. WIoU assigns a gradient gain based on the outlier degree of the anchor box, focusing on average-quality samples with moderate outliers to enhance the generalization ability of the EDC-YOLO model. The WIoU formula can be expressed as
(9)$$L_{WIoU}=r\cdot e^{\frac{{\left(x-x_{gt}\right)}^{2}+{\left(y-y_{gt}\right)}^{2}}{w_{g}^{2}+H_{g}^{2}}}\cdot L_{IoU},\beta =\frac{L_{IoU}}{\bar{L_{IoU}}},r=\frac{\beta }{\delta \alpha^{\beta -\delta }}$$
where r is the nonmonotonic focal coefficient, β is the outlier degree, $L_{IoU}$ is the loss value corresponding to the intersection and union ratio of the anchor box, and $\bar{L_{IoU}}$ is the average intersection and union ratio of the entire sample set. $x,y,x_{gt},y_{gt}$ are the dimensions of the prediction box and the target box, respectively, and $W_{g},H_{g}$ are the dimensions of the smallest closure rectangles of the prediction box and the target box. The interplay between the numerator and denominator of the outlier degree causes both overly large and overly small outlier degrees to be assigned smaller gradient gains, allowing the model to focus on average-quality samples with moderate outlier degrees. This is designed to enhance the generalization ability of the EDC-YOLO model.

### 4.2. Target Detection and Quantization Based on EDC-YOLO Modeling

#### 4.2.1. EDC-YOLO Model

A schematic representation of the locations of the C3STR, CBAM, and DConv modules within the EDC-YOLO model, relative to the number of layers, is depicted in Figure 15.

The CBAM module is positioned within the backbone network, while both the DConv and C3STR modules are integrated into the header network of EDC-YOLO. The Spatial Pyramid Pooling Facilitation (SPPF) module conducts pooling operations at various scales on the preamble feature map, extracting semantic features at multiple levels. The CBAM attention mechanism emphasizes the most critical parts of the image. The neck of the EDC-YOLO network utilizes the C3STR module to capture large-scale or global feature information, while the DConv module extracts features of varying shapes and scales. These sub-modules collectively form a Path Aggregation Network (PAN) structure, fusing feature information across different levels through bottom-up and top-down pathways, thereby enhancing the model’s feature representation capabilities. The remaining head network is responsible for the final tasks of target detection and classification.

#### 4.2.2. Experimental Environment and Parameters

The graphics card used in this study was an NVIDIA GeForce RTX 3080 with 10,239 MiB of video memory. The Python version was 3.9.19, the Torch version was 2.0.1, the CUDA version was 11.7, and the supporting CUDNN version was 8500. The EDC-YOLO network contained 301 layers and 4,328,245 network parameters, and the model’s computational capacity reached 72 GFLOPs. The model occupied 2.07 gigabytes of video memory. The model automatically adjusted the key hyperparameters through an adaptive method, as shown in Table 6.

To validate the improved performance of the EDC-YOLO model, a comparison was conducted with the lightweight YOLOv8n model from the YOLO series, using identical parameter settings for both models to ensure fairness. Additionally, the EDC-YOLO model employed the WIoU loss function, as introduced in Section 4.1.4, while the YOLOv8n model utilized the original CIoU loss function.

#### 4.2.3. Analysis of Results

Figure 16 presents the average precision–recall (P-R) curves for all prediction categories across various configurations of YOLOv8, including those enhanced by the addition of the DConv, CBAM, C3STR, and WIoU modules. Generally, if one model’s P-R curve fully encloses another’s, it indicates superior performance.

This figure demonstrates that the EDC-YOLO model, which incorporates all of the aforementioned modules, consistently outperforms the other models. Notably, the isolated addition of certain modules does not significantly enhance the detection performance for specific target categories. Instead, it is the combined effect of these modules—through mutual reinforcement and compensation for individual limitations—that leads to substantial performance improvements.

To provide a comprehensive evaluation of the model’s classification and target localization capabilities, the mean average precision (mAP50) metric is employed, with the IoU threshold set to 0.5. This metric measures the performance of models where there is significant overlap between the prediction box and the target box, as depicted in Figure 17.

According to the graph, EDC-YOLO achieves an mAP50 of 88.3%, marking a 4.4% improvement over YOLOv8n’s 83.9%. While the addition of a single module offers marginal gains, the integration of multiple modules significantly enhances the YOLOv8n model’s detection performance. Figure 18 further illustrates the training data curves for the first 50 epochs of both the YOLOv8n and EDC-YOLO models.

The loss curves for the prediction and target boxes indicate that the EDC-YOLO model with the WIoU loss function converges more quickly and achieves better loss values than the YOLOv8n model, which uses the CIoU loss function. This improvement can be attributed to the dynamic focusing mechanism of EDC-YOLO, which allows the model to focus on average-quality sample data, thereby enhancing both the training speed and generalization ability. The category training curves show minimal differences between the two models, suggesting that both are effective in classifying various types of CFRP defects. However, the mAP50 training curve reveals that once the epoch value reaches approximately 30, the mAP50 of EDC-YOLO consistently surpasses that of the YOLOv8n model, indicating superior categorization and target localization capabilities.

Figure 19 presents the detection results of the improved EDC-YOLO model for single CFRP defective targets. The upper three pairs of plots were generated by the YOLOv8n model, while the lower three pairs display the results of single-target detection using the EDC-YOLO model. For crack damage, although the YOLOv8n model can recognize the type of damage, it exhibits detection ghosting, which complicates subsequent defect processing. In contrast, the EDC-YOLO model shows generally improved category prediction probabilities, with the prediction probability for impact damage increasing from 77% to 81% and that for delamination damage from 92% to 96%, thereby enhancing the overall detection performance.

Considering the occurrence of multiple defects during the eddy current detection of CFRP in a single procedure, it is essential to evaluate the model’s effectiveness for multi-defect target detection. To this end, this study combined different types of CFRP defect images—two cracks, one impact damage, and one delamination damage—into a single image and input this into the EDC-YOLO model to simulate the multi-target detection process. The results of this detection process are shown in Figure 20.

The test results demonstrate that the EDC-YOLO multi-target detection model is capable of simultaneously identifying and localizing multiple typical CFRP defects, significantly broadening its potential applications in engineering. This model not only overcomes the limitations of traditional models, which are restricted to classifying single CFRP defects, but also quantifies the defect signal area within CFRP eddy current detection images based on localization.

This study used CFRP impact damage defects as a case study. The eddy current detection results for impact energies of 2.5 J, 3.5 J, 5.5 J, and 6 J were utilized to detect defects with the EDC-YOLO model, as illustrated in Figure 21.

By extracting the width and height parameters of the prediction box, the EDC-YOLO model’s predicted area for the CFRP impact defect region was obtained, as presented in Table 7.

The results indicate that the prediction box area of the EDC-YOLO model increases with rising impact energy, with the predicted defect area growing from 271.1 to 286.4 as the impact energy increases from 2.5 J to 3.5 J. Additionally, the damage area expands by 7.2% when the impact energy increases from 5.5 J to 6 J. These findings confirm that the EDC-YOLO model can accurately classify, identify, and quantify single-target or multi-target defects in a single pass.

### 4.3. Ablation Experiment

#### 4.3.1. Single-Module Ablation Study

To further assess the impact of the four modules—DConv, CBAM, C3STR, and WIoU—in enhancing the performance of YOLOv8n, an ablation study was conducted. This study provides metric curves for the first 50 training epochs across different scenarios, as illustrated Figure 22.

The results reveal that the EDC-YOLO model achieves superior prediction box loss compared to other models on both the training and validation sets. Moreover, the mAP50 and mAP50-95 metrics consistently remain higher during the later stages of training, while the accuracy and recall stabilize at higher levels. Regarding category prediction, the curve trends for several models are similar, with almost all models accurately categorizing the CFRP defect types. Consequently, the EDC-YOLO model significantly enhances the target identification, localization, and area labeling capabilities. The metric values for different models at the 50th epoch are presented in the following Table 8.

As shown in the table, the implementation of the WIoU module significantly reduces the prediction box loss for the YOLOv8n model on the sample training set, decreasing from 1.03 to 0.45—a reduction of 57% for the EDC-YOLO model. This reduction enhances the model’s generalization ability and convergence speed by allowing it to focus on samples of average quality. All four modules contribute to improvements in the mAP50 and mAP50-95 metrics, with EDC-YOLO demonstrating the most significant enhancement. This model excels in capturing features across different scales and shapes, leading to better detection performance for CFRP defect targets.

The accuracy of EDC-YOLO is 89.1%, which is slightly lower than the 90.2% achieved with the WIoU module and the 89.3% with C3STR, but higher than the accuracy with the DConv and CBAM modules. This discrepancy may be attributed to the large-scale nature of CFRP defects. The C3STR module, with its global self-attention mechanism, is particularly effective in identifying large-scale CFRP damage. In contrast, the DConv and CBAM modules, which are more sensitive to small- and medium-sized targets, are less effective in accurately identifying and localizing CFRP defects on a larger scale.

The FPS metric in Table 8 demonstrates the impact of incorporating additional modules into YOLOv8n. The baseline model, YOLOv8n, achieves 53 FPS, while the modified EDC-YOLO, which integrates DConv, CBAM, and other modules, operates at 42 FPS. Although this results in a reduction in the inference speed, it remains above the real-time threshold required for industrial applications. The added complexity from these modules enhances the model’s accuracy, as reflected in the improved mAP50 of 0.88327, compared to 0.83896 for YOLOv8n.

Each module affects the performance differently. For instance, DConv increases the FPS to 67 independently, yet the combination of all modules in EDC-YOLO results in a balanced trade-off, providing substantial accuracy gains while maintaining an adequate processing speed. In conclusion, while EDC-YOLO sacrifices some speed, it achieves a significant improvement in accuracy, making it a robust solution for real-time industrial detection tasks where precision is critical.

#### 4.3.2. Combined-Module Ablation Study

To further evaluate the performance impact of integrating multiple modules, we conducted additional experiments by combining two and three of the enhanced modules in EDC-YOLO. The goal of these experiments was to observe how the module combinations influence the model’s detection accuracy and performance. Three combinations were tested.

DConv + CBAM

The first combination integrated DConv and CBAM to leverage both the enhanced feature extraction from DConv and the attention mechanism provided by CBAM. This combination was expected to improve the detection precision by enhancing both the spatial and channel-wise attention, while maintaining a reasonable inference speed.

C3STR + CBAM

The second experiment combined C3STR with CBAM. C3STR improves the feature representation through optimized residual connections, and CBAM enhances the model’s attention mechanism by focusing on important features spatially and channel-wise. This combination was expected to significantly enhance the detection accuracy by improving both the structural representation and attention, particularly in complex object detection tasks.

DConv + CBAM + C3STR

Finally, we combined DConv, CBAM, and C3STR to analyze the cumulative effect of these three modules on the model performance. This combination was expected to balance enhanced feature extraction, attention mechanisms, and structural representation, aiming for a significant increase in the detection accuracy while potentially incurring a moderate trade-off in terms of the inference speed.

Table 9 highlights the performance metrics of the different module combinations, including the fully integrated EDC-YOLO model. This analysis focuses on the key metrics to evaluate the similarities, differences, and overall advantages of integrating these modules into EDC-YOLO.

The FPS values across the first three combinations (DConv + CBAM, C3STR + CBAM, DConv + CBAM + C3STR) remain consistent at 50–51 FPS, indicating that the addition of two or three modules has a minimal impact on the model’s real-time processing capabilities. This consistency suggests that these combinations are effective in maintaining high-speed inference, crucial for real-time applications such as industrial quality control or autonomous systems. The box loss values for both training and validation across all combinations are similarly close, with minor variations, indicating that all combinations perform well in terms of convergence and generalization across the training and validation datasets.

The primary difference between the combinations lies in the accuracy metrics.

DConv + CBAM delivers the best performance in terms of the mAP50 (0.88) and mAP50-95 (0.608), indicating strong overall detection performance. The combination of DConv’s enhanced convolutional features and CBAM’s attention mechanism allows this setup to excel in object localization and classification tasks.

C3STR + CBAM achieves comparable results with an mAP50 of 0.879, which is slightly lower than that of DConv + CBAM but still competitive. This combination is optimized for both feature extraction (C3STR) and spatial channel attention (CBAM), which enables precise object detection. It also has the highest recall (0.807), suggesting better performance in capturing a higher proportion of true positives, useful in tasks where minimizing missed detections is crucial.

DConv + CBAM + C3STR has the lowest mAP50 at 0.867 and mAP50-95 at 0.583, but still performs competitively. This combination demonstrates that, while adding more modules increases the model’s complexity, it may lead to diminishing returns in the detection accuracy. Nevertheless, its overall performance is still robust, especially given its ability to maintain an FPS of 50.

The fully integrated EDC-YOLO model showcases superior performance in terms of both accuracy and precision. It achieves the highest mAP50 of 0.88327, which surpasses that of all other combinations, indicating that integrating DConv, CBAM, C3STR, and WIoU results in the most effective detection model. Moreover, EDC-YOLO achieves high precision (0.8905) and competitive recall (0.79347), indicating that it can detect objects accurately without compromising its ability to correctly classify them.

However, EDC-YOLO’s FPS is slightly lower, at 42, compared to the other combinations. While this reflects a trade-off between the accuracy and processing speed, the model’s FPS remains within the acceptable range for real-time applications, particularly in scenarios where the detection accuracy is prioritized over the maximum speed.

## 5. Conclusions

This study successfully addresses the classification of several typical CFRP defects using the YOLOv8n model, demonstrating superior performance compared to visual processing models like Swin-Transformer and achieving correct classification for all test samples. The enhanced EDC-YOLO model effectively detects single-target defects by integrating deformable convolutional networks with the CBAM mechanism, thereby improving the feature extraction for small and medium-sized defects of varying shapes. Additionally, the inclusion of Swin-Transformer’s self-attention module enables the model to capture feature information on a large, even global, scale. To further refine the model’s focus on normal-quality training samples—thereby enhancing its generalization and training efficiency—a WIoU loss function is employed in the detection head of EDC-YOLO. The results demonstrate a 4.4% increase in the mAP50 with the improved EDC-YOLO, along with significantly enhanced prediction probabilities for each type of CFRP defect.

For scenarios involving multiple coexisting CFRP defects, this study shows that EDC-YOLO can accurately identify and localize multi-target defects by simulating equivalent situations. Furthermore, a quantitative analysis of the defect damage area for CFRP impact damage defects reveals that the predicted defect area by EDC-YOLO increases with rising impact energy, indicating the model’s capability to effectively learn and interpret the features and patterns of impact damage.

## Figures and Tables

**Figure 1 sensors-24-06753-f001:**
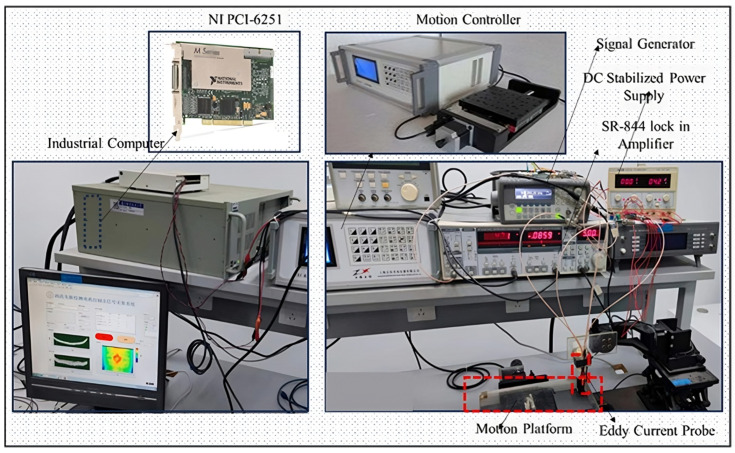
Eddy current system. The region enclosed by the red box indicates the area designated for placing the CFRP defective specimens.

**Figure 2 sensors-24-06753-f002:**
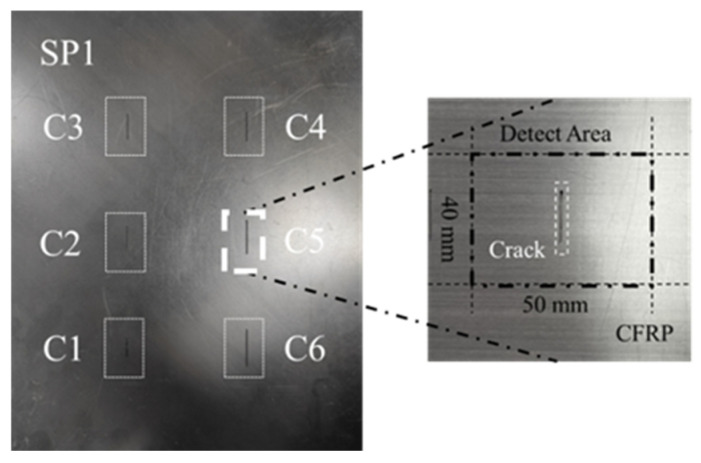
CFRP specimen 1 with crack defects.

**Figure 3 sensors-24-06753-f003:**
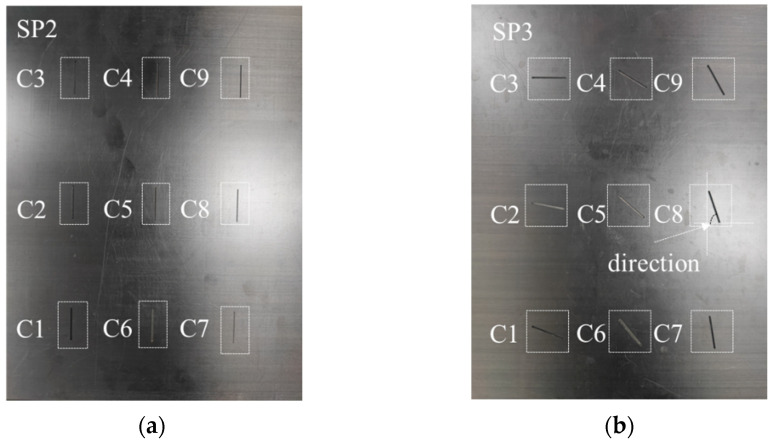
CFRP specimens 2 (**a**) and 3 (**b**) with crack defects.

**Figure 4 sensors-24-06753-f004:**
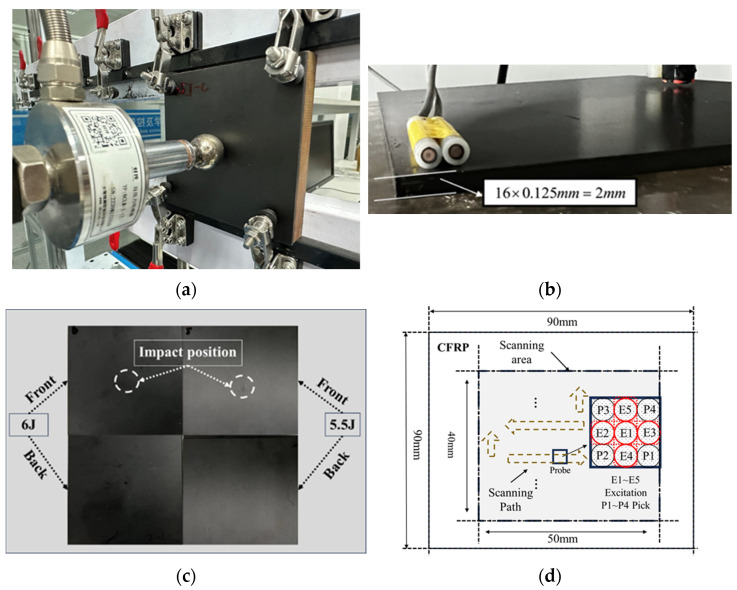
Drop hammer impact test equipment and impact damage detection schematic diagram, the detection trajectory of the probe is represented by the dashed arrow. (**a**) Impact equipment; (**b**) schematic diagram of plywood thickness direction; (**c**) CFRP defective specimens under two impact energies; (**d**) schematic diagram of eddy current detection.

**Figure 5 sensors-24-06753-f005:**
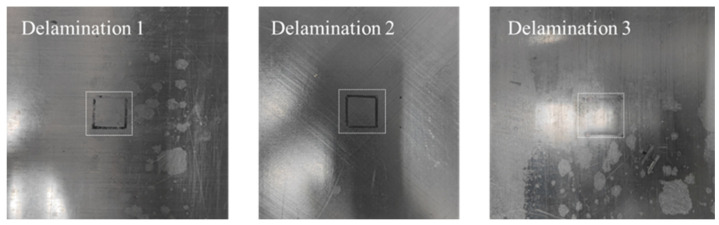
CFRP specimens with delamination defects.

**Figure 6 sensors-24-06753-f006:**
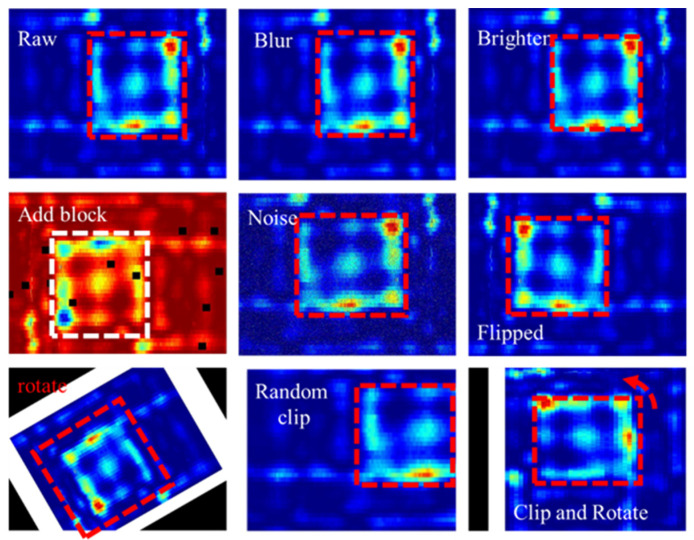
Image augmentation. The areas enclosed by the red dashed lines in the figure indicate the locations of the delamination defects.

**Figure 7 sensors-24-06753-f007:**
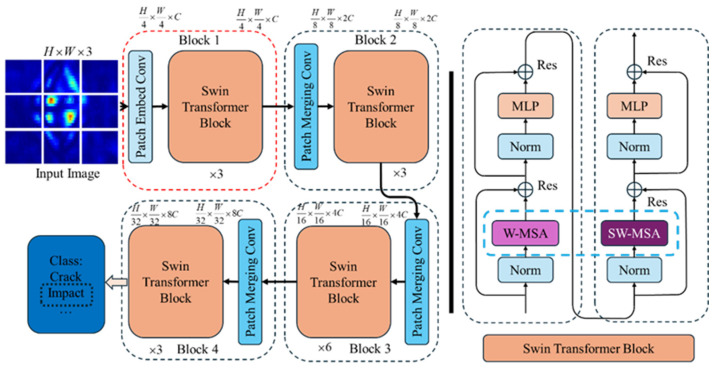
Swin-Transformer model.

**Figure 8 sensors-24-06753-f008:**
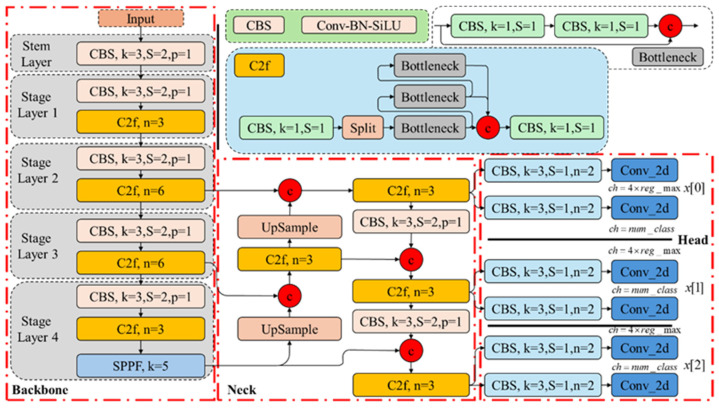
YOLOv8.

**Figure 9 sensors-24-06753-f009:**
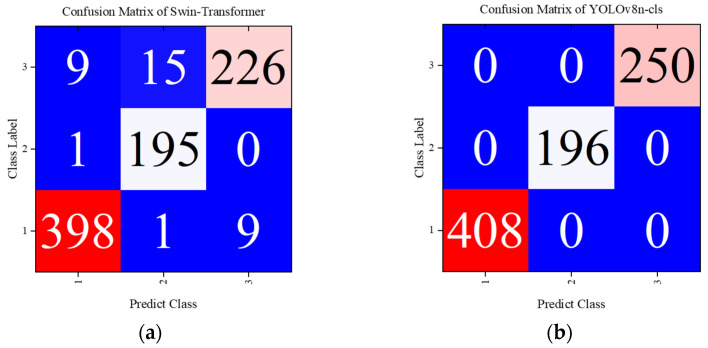
Confusion matrices of Swin-Transformer and YOLOv8. (**a**) Swin-Transformer; (**b**) YOLOv8n-cls.

**Figure 10 sensors-24-06753-f010:**
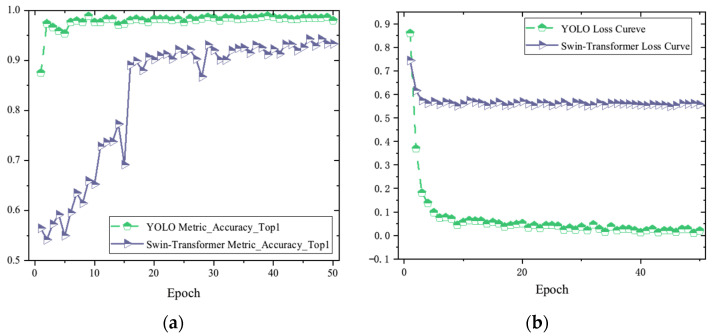
Training process. (**a**) Accuracy; (**b**) normalized loss curve.

**Figure 11 sensors-24-06753-f011:**
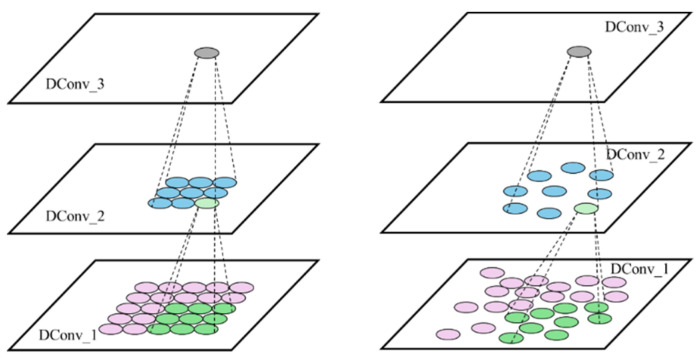
Principle of DConv.

**Figure 12 sensors-24-06753-f012:**
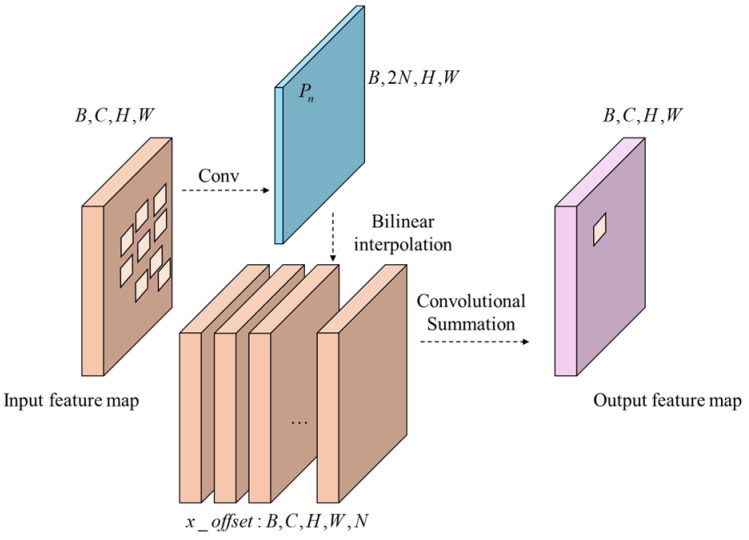
Details of DConv.

**Figure 13 sensors-24-06753-f013:**
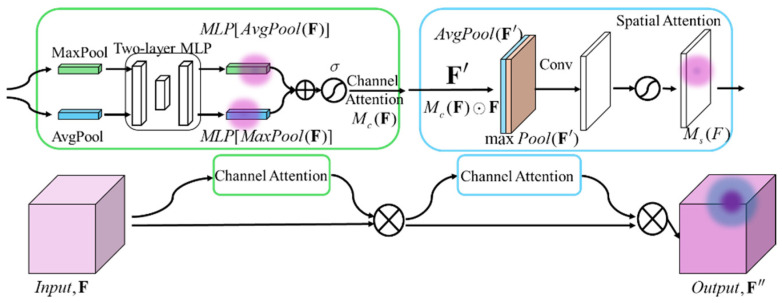
CBAM structure.

**Figure 14 sensors-24-06753-f014:**
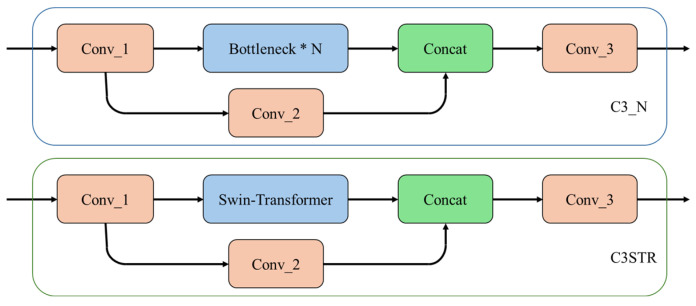
C3STR.

**Figure 15 sensors-24-06753-f015:**
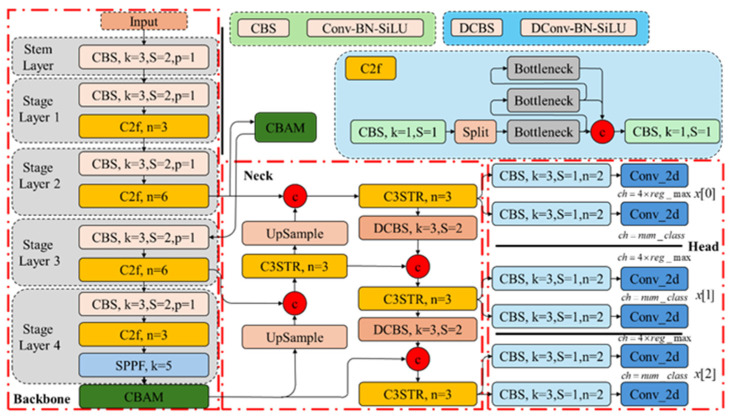
EDC-YOLO.

**Figure 16 sensors-24-06753-f016:**
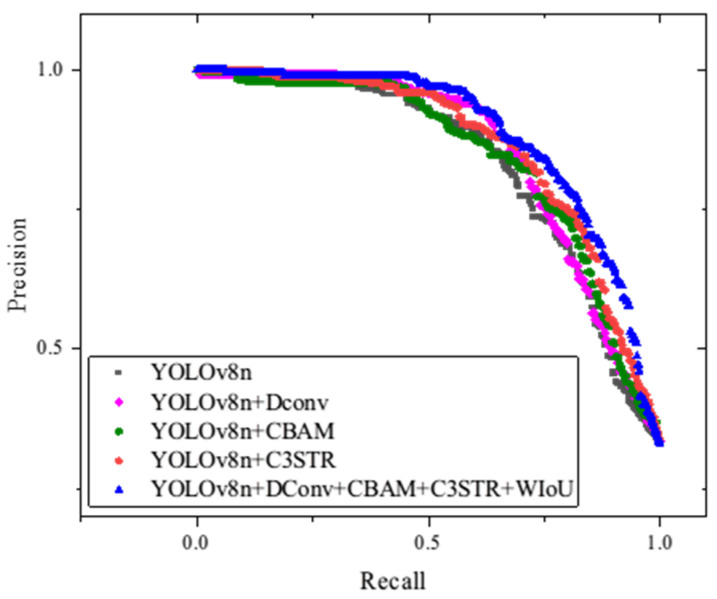
PR curves of different models.

**Figure 17 sensors-24-06753-f017:**
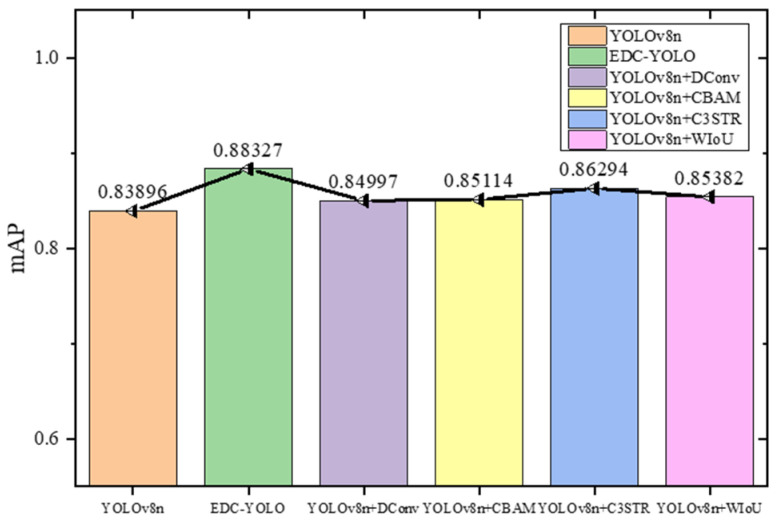
mAP50.

**Figure 18 sensors-24-06753-f018:**
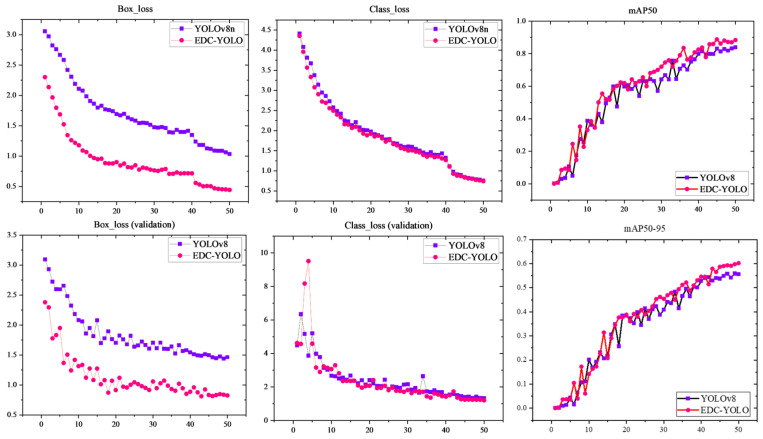
Training curves.

**Figure 19 sensors-24-06753-f019:**
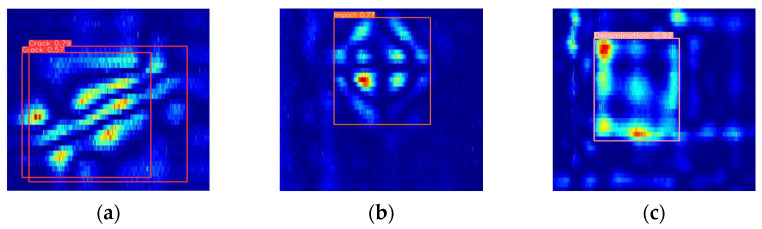
Single-defect detection of YOLOv8n and EDC-YOLO. (**a**) YOLOv8n_Crack; (**b**) YOLOv8n_Impact; (**c**) YOLOv8_Delamination; (**d**) EDC-YOLO_Crack; (**e**) EDC-YOLO_Impact; (**f**) EDC-YOLO_Delamination.

**Figure 20 sensors-24-06753-f020:**
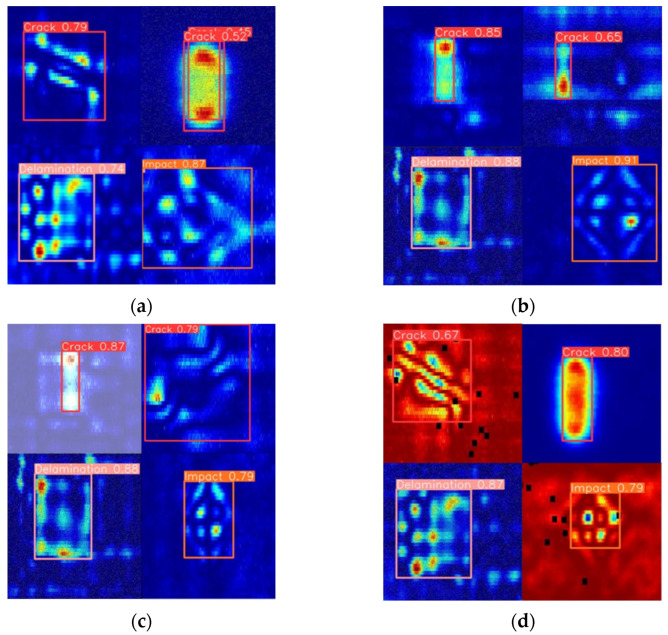
Multi-defects in a single detection image. (**a**) Multi-Objection 1; (**b**) Multi-Objection 2; (**c**) Multi-Objection 3; (**d**) Multi-Objection 4.

**Figure 21 sensors-24-06753-f021:**
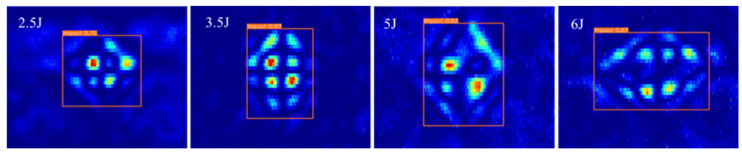
Impact defect quantization.

**Figure 22 sensors-24-06753-f022:**
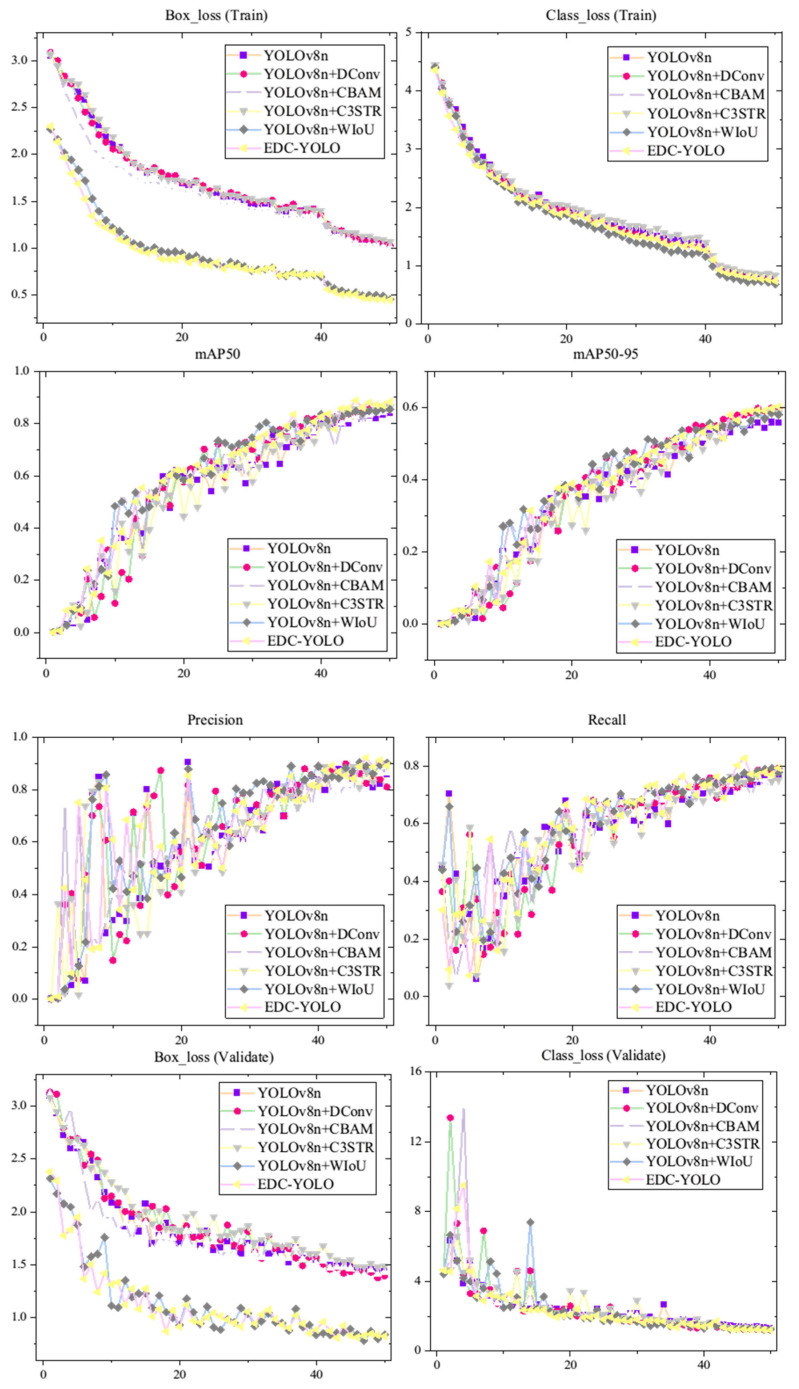
Results of ablation experiments.

**Table 1 sensors-24-06753-t001:** Crack parameters of CFRP specimen 1.

Crack No.	Length (mm)	Width (mm)	Depth (mm)	Direction (deg)
C1	11	1	1	90
C2	13			
C3	15			
C4	17			
C5	19			
C6	21			

**Table 2 sensors-24-06753-t002:** Crack parameters of CFRP specimen 2.

Crack No.	Crack Length (mm)	Crack Width (mm)	Crack Depth (mm)	Direction (deg)
C1	20	1.5	0.3	90
C2			0.6	
C3			0.9	
C4			1.2	
C5			1.5	
C6			1.8	
C7			2.1	
C8			2.4	
C9			2.7	

**Table 3 sensors-24-06753-t003:** Crack parameters of CFRP specimen 3.

Crack No.	Crack Length (mm)	Crack Width (mm)	Crack Depth (mm)	Direction (deg)
C1	20	1.5	0.3	90
C2			0.6	
C3			0.9	
C4			1.2	
C5			1.5	
C6			1.8	
C7			2.1	
C8			2.4	
C9			2.7	

**Table 4 sensors-24-06753-t004:** Parameters of delamination damage defects.

Specimen No.	Layering Sequence	Area of Delamination (mm^2^)	Depth of Delamination (mm)
1	${\left[0\right]}_{24}$	20×20	0.25
2	${\left[-45/0/45/90\right]}_{6}$		0.5
3	${\left[0/90\right]}_{12}$		0.75

**Table 5 sensors-24-06753-t005:** Multiclassification metrics of models.

Model	Macro-P	Macro-R	Macro-F1	Kappa Coefficient
Vi-T	0.737	0.651	0.691	0.535
SWT	0.954	0.958	0.956	0.935
PRI-SWT	0.952	0.938	0.945	0.924
YOLOv8n-cls	1	1	1	1

**Table 6 sensors-24-06753-t006:** Hyperparameters.

Hyperparameter	Value
Input Size	640×640
Initial Learning Rate	0.001429
Momentum	0.9
Batch Size	10
Epochs	50

**Table 7 sensors-24-06753-t007:** Area for different impact energies.

Energy (J)	2.5	3.5	5.5	6
Area (mm^2^)	271.1	286.4	405.7	435

**Table 8 sensors-24-06753-t008:** Metrics of single modules.

Model	YOLOv8n	DConv	CBAM	C3STR	WIoU	EDC-YOLO
Box loss (Train)	1.0337	1.0266	1.0031	1.0692	0.45391	0.44287
Box loss (Validate)	1.4638	1.3951	1.4287	1.4877	0.83876	0.8248
mAP50	0.83896	0.84997	0.85114	0.86291	0.85382	0.88327
mAP50-95	0.55683	0.59875	0.58223	0.58546	0.58012	0.60248
Precision	0.8579	0.81009	0.86498	0.89262	0.90194	0.8905
Recall	0.7595	0.79232	0.7806	0.75226	0.77833	0.79347
FPS	53	67	50	45	51	42

**Table 9 sensors-24-06753-t009:** Metrics of combined modules.

Model	DConv + CBAM	C3STR + CBAM	DConv + CBAM + C3STR	EDC-YOLO
Box loss (Train)	0.42	0.453	0.448	0.44287
Box loss (Validate)	0.836	0.886	0.817	0.8248
mAP50	0.88	0.879	0.867	0.88327
mAP50-95	0.608	0.6	0.583	0.60248
Precision	0.893	0.868	0.869	0.8905
Recall	0.791	0.807	0.775	0.79347
FPS	51	51	50	42

## Data Availability

All relevant data are within the paper.

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
