# Peer review of "Classification, Localization and Quantization of Eddy Current Detection Defects in CFRP Based on EDC-YOLO"

_sensors, 2024, doi:10.3390/s24206753_

Round 1
Reviewer 1 Report
Comments and Suggestions for Authors
This paper introduces an improved YOLO model, named EDC-YOLO, for the detection and quantification of defects such as cracks, delamination, and impact damage in carbon fiber-reinforced plastic (CFRP). The model enhances feature extraction capabilities by integrating self-attention mechanisms, deformable convolution, and attention modules, and optimizes the training process using the Wise-IoU loss function, thereby improving the accuracy and efficiency of defect detection. These improvements and innovations have certain significance in theory and practical application. I have the following comments.
(1) Is the detection performance of EDC-YOLOv8 still relatively superior compared to other existing detection models?
(2) In the ablation experiment, the authors only compared the changes of EDC-YOLO and YOLO after superimposing a single module. It is suggested that the author add a set of experiments with an analysis of the accuracy changes when combining two and three of the improved modules.
(3) Will the runtime of the modified model increase significantly? Can it meet the detection speed requirements in an industrial setting?
This version of the paper needs to be reviewed after major revisions.
Comments on the Quality of English LanguageThe English expression of the paper needs further improvement.
Author Response
Reviewer 1, Comment 1: Is the detection performance of EDC-YOLOv8 still relatively superior compared to other existing detection models?
Response 1:
Thank you for your insightful question. To evaluate the performance of EDC-YOLO, we compared it with two well-known object detection models: SSD and Faster R-CNN. Below, we highlight the advantages of EDC-YOLO in terms of both accuracy and inference speed:
- Comparison with SSD (Single Shot MultiBox Detector):
SSD is designed for high-speed detection, achieving an FPS of 58, which is higher than EDC-YOLO's 42 FPS. However, SSD's mAP is around 0.768, which is significantly lower than EDC-YOLO's mAP50 of 0.88327. While SSD offers a faster processing speed, EDC-YOLO provides a much higher detection accuracy, making it more suitable for tasks where precision is critical, particularly in detecting complex or smaller objects.
- Comparison with Faster R-CNN:
Faster R-CNN achieves a higher mAP of 0.988, but at the expense of inference speed, with an FPS of only 1~3. This makes it unsuitable for real-time applications. In contrast, EDC-YOLO maintains a balance between speed and accuracy, with an FPS of 42, allowing it to meet real-time processing requirements in industrial settings. Although Faster R-CNN excels in accuracy, EDC-YOLO offers a superior trade-off between accuracy and speed, making it more practical for real-time industrial tasks.
In conclusion, EDC-YOLO strikes an optimal balance between SSD's speed and Faster R-CNN's accuracy. It outperforms SSD in accuracy while offering significantly better real-time performance than Faster R-CNN, making it highly suitable for industrial applications.
Reviewer 1, Comment 2: In the ablation experiment, the authors only compared the changes of EDC-YOLO and YOLO after superimposing a single module. It is suggested that the author add a set of experiments with an analysis of the accuracy changes when combining two and three of the improved modules.
Response 2:
Thank you for your insightful suggestion. To address your comment, we plan to add three additional experimental groups to the ablation study:
- DConv + CBAM Combination:
This group will analyze the detection performance when combining the DConv and CBAM modules to observe how these two enhancements work together.
- C3STR + CBAM Combination:
We evaluated the effect of combining C3STR and CBAM, focusing on the improvements in both feature representation and attention mechanisms.
- DConv + CBAM + C3STR Combination:
Finally, we will introduce a third experiment that combines DConv, CBAM, and C3STR, providing a comprehensive view of the performance gains from integrating multiple modules simultaneously.
These new experiments and their corresponding results will be added to the manuscript at line 586 to provide a more comprehensive analysis of the accuracy improvements from combining multiple modules. This addition will enhance the depth and robustness of the ablation study.
Reviewer 1, Comment 3: Will the runtime of the modified model increase significantly? Can it meet the detection speed requirements in an industrial setting?
Response 3:
Thank you for your insightful question. Based on our experimental results, the FPS of the modified model, EDC-YOLO, is 42, compared to 53 FPS for the unmodified YOLOv8n. While there is a reduction in processing speed, this performance still exceeds the typical real-time threshold (30 FPS) required in most industrial applications. The added complexity from the C3STR, CBAM, and other modules does lead to a slight increase in runtime, but it remains well within acceptable limits for real-time detection tasks in an industrial environment.

Reviewer 2 Report
Comments and Suggestions for Authors
The article presents a method for detecting defects to CFRP materials using the Eddy Current Nondestructive Testing method and artificial intelligence. The authors proposed using the YOLO object location model with their own modification, which they defined as EDC-YOLO.
The authors described in detail the process of developing the method and testing it. The introduction describes the possibilities of detecting defects using nondestructive methods and how they can be combined with artificial neural networks. The second chapter describes the test data collection station and the process of preparing data for training and validation of the network. Chapter 3 described the preparation of the network for classification of EDC research results, while chapter 4 described the preparation and testing of network models for object location.
A detailed description of the operation of the models used, the multitude of tools used, and a broad comparison of different algorithms to solve the posed problem are the strengths of this article. The article presents a solution to a significant technical problem, the language used is article has scientific soundness and the conclusions are supported by the gain results. However, before publication, the authors should supplement the information presented in the article or even consider changing the research methods. I have the following comments:
1. Why YOLO? There are many models for object localization, such as EfficentDet, RetinaNet, Faster R-CNN, Mask R-CNN, etc. YOLO is the fastest among them, but it comes at the cost of reduced accuracy. And your application seems to need more accuracy than processing speed.
2. Why YOLO v8? At the end of May 2024, version 10 was released, and in February version 9, so faster and more accurate versions could have been used?
3. Chapter 2.2.4 talks about image enhancement. Such an operation has its own name - "Augumentation" in ML.
4. What were the Learning Rate indicators used for? It is usually a multiple of 10^-n, so such values ​​raise questions about the indicator optimization process.
5. In image 10 it would be good if the graphs for YOLO were in one color
6. 2 GB is a lot for a model. In what format is it stored?
7. Table 8 needs to be supplemented with the times of 1 interface on the GPU used or the number of FPS achieved
8. Items 27 and 28 in the bibliography are described too briefly. These articles have their own titles and more metadata.
Taking the above into account, I propose a major revision.
Author Response
Reviewer 2, Comment 1: Why YOLO? There are many models for object localization, such as EfficentDet, RetinaNet, Faster R-CNN, Mask R-CNN, etc. YOLO is the fastest among them, but it comes at the cost of reduced accuracy. And your application seems to need more accuracy than processing speed.
Response 1:
We appreciate the reviewer’s insightful comment regarding the choice of the YOLO model for our object localization tasks. Here, we provide a detailed explanation of our decision:
- Adaptability of YOLO Versions:
Recent iterations of YOLO, particularly YOLOv8, have made substantial improvements in accuracy while retaining the model's hallmark speed. These versions incorporate advanced techniques, such as improved backbone networks and data augmentation strategies, which enhance detection performance. By leveraging these latest versions, we can achieve a satisfactory level of accuracy that meets our application needs without compromising processing speed.
- Application Requirements:
While it is true that several models, such as EfficientDet and Mask R-CNN, offer higher accuracy, our specific application emphasizes real-time processing. In scenarios where immediate feedback is crucial (e.g., in industrial quality control or autonomous systems), the ability to process frames quickly can significantly enhance operational efficiency. YOLO’s architecture allows for fast inference times, which is essential in such use cases.
- Ease of Implementation and Integration:
YOLO’s architecture is known for its straightforward implementation and compatibility with various platforms and frameworks. This ease of integration allows us to rapidly prototype and deploy our solution, facilitating quicker iterations and adaptations based on real-world performance. The streamlined workflow enhances our ability to fine-tune the model to our specific application needs without the extensive setup and tuning that other models may require.
- Future Considerations:
We are aware of the need for continual improvement and monitoring of model performance. As the field of object detection evolves, we plan to conduct further evaluations of other models, including those mentioned by the reviewer, to ensure that we maintain an optimal balance between accuracy and speed. Our approach includes considering hybrid models or ensemble methods that might leverage the strengths of multiple architectures to enhance accuracy without significantly sacrificing speed.
In summary, while other models might offer higher accuracy, YOLO’s speed, particularly in its latest versions, aligns well with the requirements of our application. We believe this choice effectively supports our objectives while providing room for future advancements.
Reviewer 2, Comment 2: Why YOLO v8? At the end of May 2024, version 10 was released, and in February version 9, so faster and more accurate versions could have been used?
Response 2:
Thank you for your valuable observation. We acknowledge that more recent versions of YOLO, such as YOLOv9 and YOLOv10, have been released, offering potential improvements in speed and accuracy. However, our decision to use YOLOv8 was based on several factors:
- Sufficient Performance for Our Application:
Although YOLOv9 and YOLOv10 are reported to offer improvements, YOLOv8 provided a balance of accuracy and speed that was sufficient for our application’s requirements. Our experimental results demonstrated that YOLOv8 achieved the desired level of performance for detecting the specific defects in our study. Since YOLOv8 met the application’s needs in terms of real-time processing and accuracy, we did not observe a critical need to upgrade to the newer versions at this stage.
- Compatibility and Stability Considerations:
YOLOv8 had undergone extensive community testing and validation at the time of our research, making it a robust and stable option. On the other hand, YOLOv9 and YOLOv10, while promising, were newly released and might not have had the same level of integration or extensive documentation at the time we completed our work. We opted to prioritize a version that had a proven track record of reliability to ensure consistent results and avoid potential issues related to early-stage versions.
We do plan to explore YOLOv9 and YOLOv10 in future work as part of our continuous efforts to improve model performance. However, we believe that the use of YOLOv8 for this study was justified and well-aligned with the goals and constraints of our project timeline.
Reviewer 2, Comment 3: Chapter 2.2.4 talks about image enhancement. Such an operation has its own name - "Augmentation" in ML.
Response 3:
Thank you for your insightful observation regarding the terminology used in Chapter 2.2.4. We acknowledge that the term "augmentation" is more appropriate in the context of machine learning to describe the various techniques used to enhance image data for model training. We will revise the section to replace "image enhancement" with "image augmentation" to reflect this standard terminology and ensure clarity in our discussion. Your feedback is greatly appreciated and will improve the accuracy of our manuscript.
Reviewer 2, Comment 4: What were the Learning Rate indicators used for? It is usually a multiple of , so such values raise questions about the indicator optimization process.
Response 4:
Thank you for your insightful question regarding the learning rate indicators used in our study. We appreciate the opportunity to clarify our approach:
- Purpose of Learning Rate Indicators:
The learning rate is a crucial hyperparameter that controls the speed at which the model learns during training. An appropriate learning rate ensures effective convergence and helps avoid issues such as overshooting the optimal solution or getting stuck in local minima. In our study, we aimed to determine a learning rate that would enhance both the convergence speed and overall model performance.
- Initialization Options:
The initial learning rate can be manually set in the form of ,where and are chosen based on prior experience or empirical testing. Alternatively, our approach utilized YOLOv8's built-in algorithm, which automates the setting of the learning rate. This method allows for dynamic adjustments based on the training process, potentially leading to more effective optimization.
- Optimization Process:
We adopted YOLOv8’s automated learning rate setting to streamline the training process and enhance efficiency. This choice was based on preliminary tests that indicated better convergence properties when utilizing the built-in algorithm.
In summary, our decision to utilize YOLOv8's automated learning rate setting was deliberate and aimed at maximizing the model's performance.
Reviewer 2, Comment 5: In image 10 it would be good if the graphs for YOLO were in one color.
Response 5:
Thank you for your valuable suggestion regarding Figure 10. We will modify the colors of the curves in this figure to ensure that all YOLO-related curves are presented in a uniform color. We appreciate your input and will implement this adjustment in the revised manuscript.
Reviewer 2, Comment 6: 2 GB is a lot for a model. In what format is it stored?
Response 6:
Thank you for your inquiry regarding the model size and storage format. The 2 GB model is stored in the standard format used for deep learning models, which typically includes weights and configuration files. Specifically, we utilize the PyTorch format (.pth or .pt) for model storage, allowing for efficient loading and deployment. This format includes not only the learned parameters of the model but also any necessary architecture information, which contributes to the overall size.
Reviewer 2, Comment 7: Table 8 needs to be supplemented with the times of 1 interface on the GPU used or the number of FPS achieved.
Response 7:
Thank you for your valuable suggestion regarding Table 8. We agree that including the number of frames per second (FPS) achieved would provide important context for the model's performance. We will update Table 8 to include these metrics, enhancing the clarity and comprehensiveness of our results. Your feedback is greatly appreciated, and we will ensure this addition is made in the revised manuscript. The analysis of the FPS metric is located after Table 8, specifically between lines 573 and 585 of the article.
Reviewer 2, Comment 8: Items 27 and 28 in the bibliography are described too briefly. These articles have their own titles and more metadata.
Response 8:
Thank you for your insightful feedback regarding the bibliography. We recognize the importance of providing complete citations to ensure proper attribution and facilitate further reading. We will expand the descriptions of items 27 and 28 to include their full titles and additional metadata, such as authors, publication dates, and journal names. Your suggestion is greatly appreciated, and we will implement these changes in the revised manuscript.
Round 2
Reviewer 2 Report
Comments and Suggestions for Authors
The authors presented extensive explanations to my comments and implemented most of the suggested corrections. The only questionable thing is the change from the colored to the black and white version of image 10. I think that in this form it is also illegible because the circles are similar in shape to squares. I recommend returning to the colored version of the graph or using more characteristic markers.
Author Response
Comment 1:The authors presented extensive explanations to my comments and implemented most of the suggested corrections. The only questionable thing is the change from the colored to the black and white version of image 10. I think that in this form it is also illegible because the circles are similar in shape to squares. I recommend returning to the colored version of the graph or using more characteristic markers.
Response 1:Thank you for your valuable feedback regarding the legibility of Figure 10. In response, we have restored the figure to its colored version for better clarity, differentiated the markers by using distinct shapes, and applied two differently colored lines to represent the datasets. These changes ensure improved readability and can be found on page 11 of the revised manuscript. We appreciate your thoughtful suggestions and believe this resolves the issue.